# L-Carnitine Tartrate Supplementation for 5 Weeks Improves Exercise Recovery in Men and Women: A Randomized, Double-Blind, Placebo-Controlled Trial

**DOI:** 10.3390/nu13103432

**Published:** 2021-09-28

**Authors:** Matthew Stefan, Matthew Sharp, Raad Gheith, Ryan Lowery, Charlie Ottinger, Jacob Wilson, Shane Durkee, Aouatef Bellamine

**Affiliations:** 1Applied Science & Performance Institute, Research Division, Tampa, FL 33607, USA; msharp@theaspi.com (M.S.); rgheith@theaspi.com (R.G.); rlowery@theaspi.com (R.L.); cottinger@theaspi.com (C.O.); jwilson@theaspi.com (J.W.); 2Lonza Consumer Health Inc., Morristown, NJ 07960, USA; shane.durkee@lonza.com

**Keywords:** carnitine, recovery, fatigue, exercise, L-carnitine tartrate, muscle damage antioxidant, muscle strength, muscle power

## Abstract

L-carnitine tartrate has been shown to improve relatively short-term recovery among athletes. However, there is a lack of research on the longer-term effects in the general population. Objective: The primary objectives of this randomized double-blind, placebo-controlled trial were to evaluate the effects of daily L-carnitine tartrate supplementation for 5 weeks on recovery and fatigue. Method: In this study, eighty participants, 21- to 65-years-old, were recruited. Participants were split into two groups of forty participants each, a placebo, and a L-carnitine Tartrate group. Seventy-three participants completed a maintenance exercise training program that culminated in a high-volume exercise challenge. Results: Compared to placebo, L-carnitine tartrate supplementation was able to improve perceived recovery and soreness (*p* = 0.021), and lower serum creatine kinase (*p* = 0.016). In addition, L-carnitine tartrate supplementation was able to blunt declines in strength and power compared to placebo following an exercise challenge. Two sub-analyses indicated that these results were independent of gender and age. Interestingly, serum superoxide dismutase levels increased significantly among those supplementing with L-carnitine tartrate. Conclusions: These findings agree with previous observations among healthy adult subjects and demonstrate that L-carnitine tartrate supplementation beyond 35 days is beneficial for improving recovery and reducing fatigue following exercise across gender and age.

## 1. Introduction

L-carnitine is a quaternary amine that plays a vital role in energy generation by interacting with fatty acids [1]. While the mammalian body can synthesize L-carnitine from lysine and methionine, most of our daily intake of L-carnitine comes from the diet, with red meat being an abundant source [1]. Under certain circumstances, endogenous synthesis and dietary uptake may not be sufficient and supplementation may be required [2]. L-carnitine is responsible for shuttling fatty acids from the cytosol into the mitochondria for fatty acid β-oxidation and energy production [2]. As such, L-carnitine was shown to support mitochondrial function [3]. L-carnitine is primarily stored in skeletal and cardiac muscles because of their high mitochondrial density and high energy demand [4]. Because of L-carnitine’s role in the energy-generating processes, it was shown to be an important player in muscular performance, development, recovery, and physical exercise [5,6,7].

Exercise-induced muscle damage, and the subsequent pain, can have a significant impact on exercise performance by limiting training activity and decreasing the quality of life [8]. Supplementation with L-carnitine was shown to mitigate muscle damage and to have a positive effect on recovery from exercise [3]. Previous research summarized several mechanisms of action for these effects [3]. One such plausible mechanism is L-carnitine’s ability to prevent exercise-induced muscle degradation [9] by improving protein signaling as well as mitigating leakage of cytosolic proteins from the muscle cell [10,11]. Volek and colleagues [5] found that circulating plasma myoglobin and creatine kinase (CK), both markers of muscle damage, were significantly attenuated, and that malondialdehyde, a marker of oxidative stress, returned to resting values at a faster rate with L-carnitine supplementation compared to placebo. A separate study found that L-carnitine supplementation has been shown to significantly attenuate other muscle damage markers such as hypoxanthine, xanthine oxidase, malondialdehyde, CK, myoglobin, and perceived muscle soreness [12]. Lastly, L-carnitine supplementation increased insulin-like growth factor-binding protein 3 and skeletal muscle androgen receptors when combined with exercise. These factors are involved in protein synthesis [6,10].

Strenuous exercise promotes a buildup of metabolites, leading to the formation of reactive oxygen species (ROS), which may contribute to additional muscle damage [13]. Antioxidants are known to mitigate and prevent the formation of ROS [14]. Interestingly, L-carnitine has been shown to increase indicators of antioxidant activity such as superoxide dismutase (SOD), glutathione peroxidase, catalase, and total antioxidant capacity over placebo [15]. Subsequently, subjects who are supplemented with L-carnitine for two weeks had a reduction in lipid peroxidation (based on the thiobarbituric acid-reactive substance method) and muscle damage markers (malondialdehyde, creatine kinase, and lactate dehydrogenase) with a concurrent increase in total antioxidant capacity [16].

L-carnitine tartrate supplementation was demonstrated to have attenuated biochemical markers of purine metabolism (hypoxanthine and xanthine oxidase), free radical formation (malondialdehyde), muscle tissue disruption (myoglobin and creatine kinase), and muscle soreness after physical exertion in the short (≤1 week) and medium terms (28 days) [12] during strenuous exercise; however, its effects on recovery beyond 28 days with moderate exercise have yet to be investigated. Therefore, the primary objective of this randomized, double-blinded, placebo-controlled study was to investigate the effects of supplementation of L-carnitine tartrate beyond 28 days, among 80 men and women aged 21–65-year-old, on recovery and fatigue after an exercise challenge at the end of the supplementation, by assessing perceived muscle soreness and serum CK, indicators of muscle damage. Muscle strength based on an isometric mid-thigh pull, muscle power from a countermovement jump (CMJ) and a squat jump (SJ), as well as antioxidant capacity based on serum SOD levels were evaluated as secondary outcomes. Lastly, data were stratified by age and gender in a post-hoc analysis.

## 2. Materials and Methods

### 2.1. Subjects

Subjects were recruited by word of mouth, email contact, and direct contact from around the Tampa Bay area. A total of 80 healthy male and female subjects, aged 21- to 65-years, were recruited based on their activity level (subjects were presumed to participate in moderate physical activity (resistance training activity or cardiovascular training activity including sports) at least 3 days per week where moderate activity was classified as greater than 50% of their heart rate max for 30 min, 3 days·week^−1^) and enrolled in the study. Subjects were excluded from the study if they: had a body mass index (BMI) of ≥30 kg/m^2^; presented any evidence of cardiovascular, metabolic, or endocrine diseases; had recently undergone surgery that affects digestion and absorption; smoked; drank heavily (>7 or >14 drinks per week for women and men, respectively); were women who were pregnant or planning to be pregnant; were taking medication to control blood pressure, lipids, and blood glucose; or had taken anabolic-androgenic steroids. In addition, individuals who used antioxidant supplements, nonsteroidal anti-inflammatory drugs (NSAIDS), nutritional supplements known to support recovery, immune function or muscle mass accreditation, or prescription medications, were also excluded. The study was divided into 2 cohorts of 40 subjects each (Figure 1). Table 1 provides demographic details of the 73 subjects who completed the study (Table 1). The seven subjects who did not complete the study failed to attend follow-up testing due to time constraints, work, or family responsibilities. Prior to engaging in any study procedures, subjects signed a written informed consent for participation. The protocol was approved by an external Institutional Review Board (IntegReview; Austin, TX, USA, Protocol #0220).

### 2.2. Study Protocol

This was a randomized, double-blind, placebo-controlled, parallel trial. Subjects underwent baseline testing (Pre), which included: salivary measures; drawing of blood for serum analysis of muscle damage and oxidative stress markers; maximal isometric strength; maximal muscle power; body composition analysis; and perceptual recovery measures. Immediately following baseline testing, subjects were stratified in quartiles by BMI and subjects within each quartile were randomly divided into the L-carnitine supplementation group or the placebo group using a random number generator (randomizer.org, Randomness and Integrity Services Ltd.; Dublin, Ireland). Thereafter, subjects were given their respective supplementation (Carnipure^®^ carnitine tartrate (CAR), Lonza Consumer Health Inc.; Morristown, NJ, USA; or Microcrystalline Cellulose-Based Placebo (PLA)). Capsules were manufactured by Liquidcapsule Manufacturing LLC, Tampa, FL, USA. Capsules were designed to fit 3 g of Carnipure^®^ carnitine tartrate in 3 total capsules, yielding 2 g of elemental L-carnitine or 3 g of placebo-matching capsules. Chemical and microbial stabilities were performed on the finished capsules by Liquidcapsule Manufacturing, LLC. Both CAR and PLA capsules were stored in visually identical capsules and containers. Subjects were instructed to consume three capsules a day for the duration of the study, either 30 min prior to exercise or with the first meal of the day on non-exercise days. Participants engaged in a moderate 5-week exercise training program twice per week with the final training session on the fifth week of supplementation consisting of a supervised lower-body exercise challenge specifically constructed to induce muscle damage. Subjects continued to supplement for two days after completing the exercise challenge. On the final training session of week 5 (Wk5-Pre), and approximately 48 h post-training (Wk5-Post), subjects were retested in a manner identical to baseline measures with the exception of body composition analysis, which was repeated at Wk5-Pre, rather than Wk5-Post, to avoid variation from exercise-induce edema following the exercise challenge. Study procedures are further described below.

### 2.3. Serum Biomarkers of Muscle Damage and Antioxidant Status

Subjects were fasted for 10 h prior to venous blood being extracted by venipuncture of the antecubital vein using a 21-gauge syringe and collected into an 8 mL collection tube containing serum separation gel (Vacuette^®^, Refence # 155071P; Greiner-Bio-One, Kremsmünster, Austria) by a certified phlebotomist. Following collection, blood samples were inverted 4–6 times and placed in a test tube holder at room temperature for 30 min to clot. Thereafter, blood samples were centrifuged at 2000× *g* for 15 min at 4 °C. The resulting supernatant were then aliquoted and stored at −80 °C until further analysis. Serum CK, total carnitine, and SOD were assayed via commercially available ELISA kits (ThermoFisher Scientific, Grand Island, NY, USA). Samples were thawed once and analyzed in duplicate in the same assay for each analysis to avoid compounded inter-assay variance.

### 2.4. Perceptual Measures

The perceived muscle recovery and soreness scale (PRS) is a scale from 0 to 10 [17,18]. The scale is a composite scale, directly proportional to recovery and inversely proportional to soreness. Subjects were informed that having no soreness and a high feeling of recovery is a 10, while extreme soreness and low feelings of recovery, leading to limited ability to function, is defined by a range of 0–3, adequate recovery, with some ability to function, is defined by a range of 4–7, and good recovery, with the ability to function normally, is defined by a range of 8–10. We investigated how PRS changed following intensified training on week 5 in both groups compared to Pre (true baseline values). Values were normalized to training volume on the muscle-damaging exercise session performed on week 5. The formula used was as follows:((Post Exercise PRS − True Baseline Pre)/total repetitions) × 100.(1)

### 2.5. Salivary Immunoglobulin A

Immune function was examined via salivary expression of immunoglobulin A. Salivary samples were taken using IPRO Oral Fluid Collector (OFC) Kits (Soma Bioscience; Wallingford, UK). The OFC kits collect 0.5 mL of oral fluid and contain a color changing volume adequacy indicator within the swab, giving collection times typically in the range of 20–50 s [19]. Levels of salivary immunoglobulin A (sIgA) were assessed in the morning, following an overnight fast. The samples were analyzed using an IPRO POC Lateral Flow Device (LFD), specific for IgA in an IPRO LFD Reader (Soma Bioscience, Wallingford, UK). Two drops of saliva/buffer mix from the OFC were added to the sample window of the LFD. The liquid ran the length of the test strip via lateral flow, creating a control and test line visible in the test window. Ten minutes after the sample was added, the test line intensity was measured in an IPRO LFD Reader and converted to a quantitative value.

### 2.6. Maximal Muscle Power

Maximal muscle power was assessed via the CMJ and SJ on a dual force plate platform (Leonardo Mechanograph^®^ GRFP XL; Novotec Medical GmbH, Pforzheim, Germany). The platform is composed of two symmetrical force plates that separate the platform into a left and a right half. The resonance frequency of each plate is at 150 Hz. Each plate contains four strain gauge force sensors (the whole platform, therefore, has eight force sensors). The sensors were connected to a laptop computer via a USB 2.0 connection. The signal from the force sensors were sampled at a frequency of 800 Hz and were analyzed using the Leonardo Mechanography GRFP Research Edition^®^ software (in this study version 4.2-b05.53-RES was used).

Prior to the test, subjects completed a warm-up of 10 body weight squats and two submaximal effort CMJs and SJs. Subjects were instructed to stand in a comfortable and upright position with the feet about shoulder-width apart and parallel to each other. Subjects then performed a countermovement by flexing the hips and knees. Once subjects reached a preferred countermovement depth, they explosively extended their hip, knee, and ankle joints to perform a maximal vertical jump. Subjects performed 3 hands-free, maximal effort CMJs with 30 s of rest between jumps. After completing 3 CMJs, subjects rested for 2 min and performed 3 hands-free, maximal effort SJs separated by 30 s of rest. The SJ started from a similar upright position as the CMJ, and subjects were instructed to lower into a squat position (~90° knee angle). Subjects held their squat position for approximately 3 s until a “jump” command was announced by the researcher. Immediately following the command, subjects jumped as high as possible from the squat position without reloading or further descent.

### 2.7. Maximal Isometric Muscle Strength

Each subject was tested for maximal isometric strength using an isometric mid-thigh pull (IMTP) performed in an Olympic style half rack to allow fixation of the bar at any height. Subjects were provided with liquid chalk to grip the bar. Utilizing a pronated clean grip, subjects were instructed to assume a body position similar to the second pull of the snatch and clean. The knee angle was confirmed between 125 and 135° using a hand-held goniometer and the hip angle was approximately set at 175°. Once body positioning had been stabilized, the subject was given a countdown. Minimal pre-tension was allowed to eliminate slack prior to initiation of the IMTP. Each subject performed two warm-up reps, one at 50% and one at 75% of perceived maximum effort. Thereafter, subjects completed two maximal IMTPs separated by approximately two minutes of rest. Subjects were instructed to pull fast and hard and were given strong verbal encouragement during the assessment. Force kinematics from the IMTP were collected and recorded using a linear position transducer [20].

### 2.8. Exercise Training Program

All participants completed a two-day per week training program for four weeks. The exercise program consisted of one upper-body circuit day, and one lower-body circuit day, and each day was constructed with multiple supersets. Each superset consisted of two to four exercises, and each superset was performed three to four times before progressing to the next superset. No rest was given within a superset, and participants were instructed to finish supersets as quickly as possible; however, up to two minutes were allowed between supersets. Each day consisted of either an upper-body warm-up, or a lower-body warm-up, and finally a cool down following each day. Total time commitment to the training program was about one hour per training session, for a total of two hours of training per week.

The upper body circuit day consisted of four total supersets and included the following exercises: push-ups, dips, pull-ups, inverted rows, shoulder taps, superman exercises, planks, mountain climbers, burpees, medicine ball throws, and various abdominal exercises. The lower-body circuit day consisted of 4 total supersets and included the following exercises and their different variations; squats, lunges, glute bridges, single-leg deadlifts, jumps, jumping jacks, kettlebell swings, and biking.

The fifth and final week of the training program was constructed differently than the first four weeks. On the first training day of the fifth week, participants completed an upper body circuit training day. On the second and final training session during week five of the study, and to assess the effect of L-carnitine on recovery, participants were exposed to an exercise challenge meant to induce muscle damage. The protocol consisted of a dynamic warm up, where subjects were instructed to perform rear-foot-elevated split squats. To set up the exercise, participants were instructed to place one foot on the ground, and the other foot was elevated directly behind them on a leg roller pad adapted to a squat rack. Subjects then descended downward, and then ascended upward. That counted as one repetition. Subjects were instructed to only ascend or descend in response to an audible metronome beep at a cadence of one second downward followed by one second upward. Subjects were instructed to complete as many repetitions as possible until momentary muscular failure was reached, which was defined as a) volitional muscle failure or b) failure to complete two successive repetitions at the prescribed metronome cadence. Once one leg was completed, subjects were given a rest of one minute, then instructed to switch legs, and repeated the exercise on that opposite leg. Subjects were asked to perform five sets per leg to muscular failure. Volume was tracked by counting total repetitions on the rear-foot-elevated split squats. In totality for the entire exercise program, the participants completed 10 exercise training sessions.

### 2.9. Body-Composition Analysis

Body mass was measured to the nearest 0.1 kg using a digital scale, and each subject verbally reported their height (Seca, Chino, CA, USA). Total body composition was determined by a whole-body scan on a dual-energy x-ray absorptiometry device (DXA) (Horizon A DXA System, Hologic Inc, Marlborough, MA, USA). Each subject was scanned by a certified technician, and the digital segmentation was determined via a computer algorithm. Fat-free mass, fat mass, and body fat percentage was determined for each scan. The subject was asked to wear comfortable clothing and remove any items that could attenuate the X-ray beams (jewelry, items containing wire, shoes, etc.). The subject was asked to lie in a supine position with knees and elbows extended and instructed not to move for the entire duration of the scan (approximately 5 min). The DXA has a switching-pulse system that rapidly alternates the voltage of the X-ray generator, producing two beams of high and low energies. The attenuated X-rays that have passed through the subject are measured sequentially with a detector situated on the scanning arm above the patient. An internal wheel corrects for any small fluctuations caused by this method of beam generation. The results from each scan were uploaded and accessed on a computer that was directly linked to the DXA device. Calibration of the densitometer on the DXA device was performed against a phantom provided by the manufacturing company prior to testing.

### 2.10. Adverse Events

Subjects were asked to report any adverse events on each visit in terms of incidences of headache, dizziness, nausea, vomiting, heart palpitations, trouble swallowing pills, lethargy, memory loss, cramps, chest pain, wheezing, ear pain, indigestion, blood in urine, blood in stool, ringing in ears, lethargy, swelling, and itching. Further, in the event that any severe adverse events were to have occurred, such as a life-threatening event, hospitalization, disability, or permanent damage, the investigators would have immediately informed the sponsor and stopped the study.

### 2.11. Supplement Compliance

At each successive testing time point following Pre, subjects handed in their capsule container to a blinded researcher who counted the number of remaining capsules, if any. Supplement compliance was determined as the number of capsules administered minus the number of capsules remaining multiplied by 100. The average compliance for the PLA and CAR groups over the course of the entire study were 95.8% and 97.6%, respectively.

### 2.12. Statistical Analysis

Statistical analysis was conducted on the per-protocol population using GraphPad Prism 9 (GraphPad Software, San Diego, CA, USA). Prior to carrying out inferential statistics on the 73 completers (per protocol), data were screened for outliers and normality. Visual inspection of box plots was used as a graphical method screening for outliers. Quantitatively, outliers were screened by examination of studentized residuals whereby values ±3 were considered outliers. Normality was graphically assessed through visual inspections of Q-Q plots and then confirmed through Shapiro–Wilk testing (*p* > 0.05). Following outlier and normality testing, dependent variables were scrutinized using a two-way mixed model analysis of variance (ANOVA) with group as the “between-group” factor (CAR and PLA), time as the “within-group” factor (Pre, Wk5-Pre, Wk5-Post), and subjects as a random factor. Whenever a significant F-value was obtained, post hoc testing was performed with a Bonferroni correction for multiple comparisons. For ANOVA procedures, homogeneity of variances and covariances were confirmed by Levene’s test and Box’s M test, respectively. Additionally, Mauchly’s test of sphericity was used to test the assumption of sphericity for two-way interactions. For select variables (IMTP strength and jump power), the relative delta change (Time2–Time1) was analyzed between groups with t-tests. Secondary analyses were conducted to scrutinize the effects of supplementation in males and females independently utilizing two-tailed, unpaired t-tests to compare relative delta changes. Due to the instilled statistical bias related to inherent physiological differences between males and females, genders were not compared against one another. A final exploratory analysis was conducted to investigate the effectiveness of CAR separately in younger and older individuals. In order to conduct this analysis, two-tailed, unpaired t-tests were deployed comparing relative delta changes in all primary and secondary variables between subjects aged 21 to 40 years and subjects aged 41 to 65 years. For all analyses, the alpha level was set a priori at *p* < 0.05. Data are presented as mean ± standard error unless otherwise stated.

## 3. Results

### 3.1. Primary Objectives

The trial achieved its two primary objectives. Both serum creatine kinase levels and Perceived Recovery Status and Soreness Scale results were significantly different between groups.

#### 3.1.1. Serum CK

A significant group by time interaction was detected for serum CK concentration (*p* < 0.001, Figure 2). Post-hoc analysis indicated that both groups demonstrated significantly greater increases in serum concentration levels of CK at Wk5-Post compared to Pre (CAR: 194%, mean diff = 322.3, 95% CI = 236.2 to 408.4 IU/L, *p* < 0.001; PLA: 358%, mean diff = 507.5, 95% CI = 396.3 to 618.6 IU/L, *p* < 0.001) and Wk5-Pre (CAR: 194%, mean diff = 322.1, 95% CI = 237.7 to 406.6 IU/L, *p* < 0.001; PLA: 351%, mean diff = 497.6, 95% CI = 387.1 to 608.1 IU/L, *p* < 0.001). Between-group differences were detected at Wk5-Post, whereby levels were significantly greater in PLA (28%; mean diff: 161.1; 95% CI: 26.6 to 295.6 IU/L; *p* = 0.016).

#### 3.1.2. Perceived Recovery Status and Soreness Scale (PRS)

Significant differences between groups were found in normalized PRS in which PLA demonstrated 33% greater decrements than CAR (PLA = −1.6 ± 0.8 vs. CAR = −1.2 ± 0.8; 95% CI = −0.8 to −0.1 arbitrary units (au); *p* = 0.021, Figure 3).

### 3.2. Secondary Objectives

Serum carnitine levels, antioxidant capacity, muscle power and strength and anti-inflammatory effects were assessed as secondary outcomes.

#### 3.2.1. Serum Carnitine Concentration

A significant group by time interaction was detected for serum total carnitine (*p* < 0.001, Figure 4). Post-hoc analysis indicated that the CAR demonstrated significantly higher levels of serum carnitine at Wk5-Pre (mean diff = 12.5; 95% CI = 9.4 to 15.5 uM; *p* < 0.001) and Wk5-Post (mean diff = 15.1; 95% CI = 12.1 to 18.2 uM; *p* < 0.001) relative to Pre. No significant within-group differences were found in PLA. Additionally, serum carnitine was significantly higher in the CAR group compared to the PLA group at Wk5-Pre (mean diff = 11.4; 95% CI = 6.0 to 16.9 uM; *p* < 0.001) and Wk5-Post (mean diff = 14.3; 95% CI = 8.9 to 19.8 uM; *p* < 0.001).

#### 3.2.2. Serum SOD Levels

A significant group by time interaction was detected for serum SOD (*p* = 0.001, Figure 5). Post hoc analysis revealed that levels were greater in CAR at Wk5-Pre (mean diff = 12.4; 95% CI = 6.5 to 18.2 IU/L; *p* < 0.001) and Wk5-Post (mean diff = 13.4; 95% CI = 7.6 to 19.3 IU/L; *p* < 0.001) compared to Pre. There were no significant between-group differences detected.

#### 3.2.3. SJ and CJ Relative Power Output

Significant main time effects were detected for maximum relative power output in the SJ (*p* = 0.008) and CMJ (*p* = 0.014). Post hoc testing indicated that Wk5-Post was significantly lower than Wk5-Pre for both jump types (SJ: mean diff = −0.95, 95% CI = −1.69 to −0.22 W·kg^−1^, *p* = 0.006; CMJ: mean diff = −0.90, 95% CI = −1.65 to −0.16 W·kg^−1^, *p* = 0.011). However, the relative changes in maximum relative power output (Figure 6) from Wk5-Pre to Wk5-Post were significantly lower in the PLA group compared to the CAR group for both CMJ (−0.29 ± 0.50 vs. −1.55 ± 0.31 W·kg^−1^; 95% CI: −2.44 to −0.10 W·kg^−1^; *p* = 0.036) and SJ (−0.47 ± 0.42 vs. −1.65 ± 0.29 W·kg^−1^; 95% CI: −2.22 to −0.16 W·kg^−1^; *p* = 0.024).

#### 3.2.4. Maximal Isometric Strength

There were no significant interaction or main effects detected for maximal IMTP (*p* > 0.05, Figure 7). However, the relative change from Pre to Wk5-Post was significantly lower in the PLA group compared to the CAR group (−2.62 ± 2.31 vs. 3.63 ± 2.08 kg; 95% CI: −12.40 to −0.07 kg; *p* = 0.047), as was the relative change from Wk5-Pre to Wk5-Post (−3.68 ± 1.53 vs. 2.18 ± 1.44 kg; 95% CI: −10.00 to −1.69 kg; *p* = 0.007).

#### 3.2.5. Serum C-Reactive Protein Levels

We previously reported the effects of L-carnitine on serum CRP [21] also shown in the Figure 8. A significant group by time interaction was detected for plasma CRP (*p* < 0.001). Post hoc analysis indicated that levels at Wk5-Pre in CAR were significantly lower than Pre (mean diff = −0.24; 95% CI = −0.35 to −0.12 mg/L; *p* < 0.001) and Wk5-Post (mean diff = −0.16; 95% CI = −0.27 to −0.04 mg/L; *p* = 0.003), whereas the PLA demonstrated significantly greater levels at Wk5-Post compared to Pre (mean diff = 0.24; 95% CI = 0.12 to 0.35 mg/L; *p* < 0.001) and Wk5-Pre (mean diff = 0.34; 95% CI = 0.22 to 0.45 mg/L; *p* < 0.001). Between-group differences were detected at Wk5-Post, whereby levels were significantly greater in PLA (mean diff: 0.35; 95% CI: 0.12 to 0.59 mg/L; *p* < 0.001) [21].

#### 3.2.6. Salivary Immunoglobulin A (sIgA)

A significant main time effect was detected for sIgA (*p* < 0.001). Post hoc analysis revealed that sIgA was lower than Pre at Wk5-Pre (mean diff = −54.3; 95% CI = −95.8 to −12.8 µg/mL; *p* = 0.006) and Wk5-Post (mean diff = −71.6; 95% CI = −122.1 to −21.1 µg/mL; *p* = 0.003). No difference between groups was observed (Appendix A).

#### 3.2.7. Volume Accumulation during Exercise-Induced Muscle Damage Protocol

There were no significant between- or within-group differences detected (*p* > 0.05) for repetition count in the dominant leg (D-Leg), non-dominant leg (ND-Leg), or total count during the muscle damage exercise protocol endured on Week 5, indicating that volume was controlled. Appendix A displays raw mean and standard error data for repetitions.

#### 3.2.8. Body Composition

There were no significant differences observed for any variable of body composition (*p* > 0.05). Raw mean and standard error data are displayed in Appendix A.

#### 3.2.9. Adverse Events

There was a total of 82 mild adverse events reported throughout the duration of the study. However, these adverse events did not appear to be related to supplementation and occurred at similar frequencies in both groups. No serious adverse effects were reported (death, hospitalization or emergency room visit). Occurrences of adverse events reported are listed in Table 2.

### 3.3. Data Analysis Based on Stratification by Age and Gender

Post-hoc analyses comparing the effects of L-carnitine supplementation between young (21 to 40 years old) and older (41 to 65 years old) subjects, as well as between genders, were performed.

#### 3.3.1. Subanalysis of Serum CK, SOD, and Carnitine Levels Stratified by Age and Gender

In the 21–40 y age group, the relative change of serum CK from Pre to Wk5-Post (mean diff = 186; 95% CI: 39 to 333 IU/L; *p* = 0.008) and Wk5-Pre to Wk5-Post (mean diff = 175; 95% CI: 28 to 321 IU/L; *p* = 0.014; Figure 9A) was significantly greater in the PLA group compared to the CAR group. In the 41–65 y age group, the relative change from Pre to Wk5-Post (mean diff = 183; 95% CI: 5 to 361 IU/L; *p* = 0.042) was significantly greater in the PLA group compared to the CAR group. A significant trend between groups was noted for the relative change from Wk5-Pre to Wk5-Post (mean diff = 176; 95% CI: −2.5 to 354 IU/L; *p* = 0.055).

In the 21–40 y age group, the relative change of SOD from Pre to Wk5-Pre (mean diff = 18.2; 95% CI: 6.5 to 29.8 IU/L; *p* < 0.001) and Pre to Wk5-Post (mean diff = 12.1; 95% CI: 0.5 to 23.8 IU/L; *p* = 0.039; Figure 9B) was significantly greater in the CAR group compared to the PLA group. No between-group differences were detected in the 41–65 y age group.

In both age groups, the relative change of total serum carnitine from Pre to Wk5-Post (41–65 y: mean diff = 13, 95% CI = 6 to 20 µM, *p* = 0.001; 21–40 y: mean diff = 15, 95% CI = 10 to 19 µM, *p* < 0.001; Figure 9C) and Pre to Wk5-Pre (41–65 y: mean diff = 10, 95% CI = 2 to 17 µM, *p* = 0.012; 21–40 y: mean diff = 12, 95% CI = 8 to 16 µM, *p* < 0.001) was significantly greater in the CAR group compared to the PLA group.

In females, the relative change of serum CK from Wk5-Pre to Wk5-Post (mean diff = −176; 95% CI: −299 to −54 IU/L; *p* = 0.006; Figure 9A) and Pre to Wk5-Post (mean diff = −154; 95% CI: −275 to −34 IU/L; *p* = 0.013) was significantly greater in the PLA group compared to the CAR group. In males, the relative change from Pre to Wk5-Post (mean diff = −255; 95% CI: −492 to −18 IU/L; *p* = 0.036) and Pre to Wk5-Pre (mean diff = −78; 95% CI: −154 to −3 IU/L; *p* = 0.044) was significantly greater in the PLA group compared to the CAR group.

For females, the relative change in SOD from Pre to Wk5-Post (mean diff = 14; 95% CI: 3 to 24 IU/L; *p* = 0.011; Figure 9B) and Pre to Wk5-Pre (mean diff = 12; 95% CI: 3 to 21 IU/L; *p* = 0.01) was significantly greater in the CAR group compared to the PLA group. The relative change in males from Pre to Wk5-Pre (mean diff = 11; 95% CI: 0.005 to 23 IU/L; *p* = 0.04) was significantly greater in the CAR group compared to the PLA group.

In both genders, the relative change of serum total carnitine from Pre to Wk5-Post (Males: mean diff = 18, 95% CI = 11 to 26 µM, *p* < 0.001; Females: mean diff = 12, 95% CI = 7 to 17 µM, *p* < 0.001; Figure 9C) and Pre to Wk5-Pre (Males: mean diff = 15, 95% CI = 9 to 21 µM, *p* < 0.001; Females: mean diff = 9, 95% CI = 5 to 14 µM, *p* < 0.001) was significantly greater in the CAR group compared to the PLA group.

This analysis indicates that the significant effects seen in serum CK, SOD, and carnitine are independent of age and gender.

#### 3.3.2. Performance Metrics Stratified by Age and Gender

In the 21–40 y age group, the relative change in maximal IMTP from Pre to Wk5-Post (mean diff = −13.5; 95% CI: 5.7 to 21.3 kg; *p* < 0.001) and Wk5-Pre to Wk5-Post (mean diff = −10.3; 95% CI: 2.5 to 18.2 kg; *p* = 0.005; Figure 10A) was significantly greater in the CAR group compared to the PLA group. No between-group differences were detected in the 41–65 y age group. No significant between-group difference was detected in relative change for SJ power output for either age group (Figure 10B). No significant between-group difference was detected in relative change for CMJ power output for either age group (Figure 10C).

The relative change in maximal IMTP in males from Pre to Wk5-Post was significantly greater in the CAR group compared to the PLA group (mean diff = 11.7; 95% CI: 2.9 to 26.4 kg; *p* = 0.050). In both genders, the relative change from Wk5-Pre to Wk5-Post was significantly greater in the CAR group compared to the PLA group (Males: mean diff = 8.5, 95% CI = 2.3 to 19.2, *p* = 0.049; Females: mean diff = 4.5 kg, 95% CI = 1.0 to 8.4, *p* = 0.020; Figure 10A). For the SJ, the relative change in males from Wk5-Pre to Wk5-Post was significantly greater in the CAR group compared to the PLA group (mean diff = 2.1; 95% CI: 0.2 to 4.0 W·kg^−1^; *p* = 0.029; Figure 10B). For the CMJ, the relative change in males from Wk5-Pre to Wk5-Post was significantly greater in the CAR group compared to the PLA group (mean diff = 3.1; 95% CI: 0.5 to 6.2 W·kg^−1^; *p* = 0.034; Figure 10C).

#### 3.3.3. Effects of L-Carnitine Supplementation on Female Menstruation Cycle

As an exploratory outcome, 19 female subjects were examined for their serum L-carnitine levels while on their menstrual cycle, with and without supplementation. Changes in L-carnitine levels before and during the menstruation cycle are shown in the Figure 11. Significant differences between groups were found, in which PLA demonstrated decrements in serum carnitine compared to CAR (*p* < 0.001).

## 4. Discussion

Prior research examined the effects of short- and medium-term L-carnitine supplementation on recovery from muscle-damaging exercise [3]. The current study sought to add to the existing research by investigating the effects of L-carnitine tartrate supplementation on longer-term recovery in a general adult population following an intense exercise challenge, which was preceded by a maintenance exercise program. Our primary hypothesis was that L-carnitine tartrate supplementation would improve recovery mainly by improving perceived muscle soreness and serum CK, an indicator of muscle damage. In addition, an exploratory analysis was carried out to examine whether L-carnitine tartrate would improve recovery in males and females independently. Specifically, L-carnitine tartrate supplementation was able to improve muscle recovery by improving both perceived muscle recovery and mitigating a rise in serum CK, which, in turn, prevented declines in strength and power. The findings in the present study were generally found to be consistent across both genders. Mechanistically, the ingredient appears to affect these parameters through elevating anti-oxidative and anti-inflammatory capacities based on increased levels in serum SOD and decreased CRP.

Exercise-induced muscle damage and oxidative stress can both decrease quality of life and limit activity as a result of fatigue [8]. Serum CK was selected as one of our primary measures of muscle damage and fatigue. The rationale is that cellular disturbances from exercise training are known to disrupt myofibrillar integrity, leading to leakage of byproducts from the cell into the blood serum [22]. As such, raised levels of CK are closely associated with cell damage, muscle cell disruption, and overtraining [22]. Our results demonstrated that L-carnitine tartrate supplementation for 5 weeks was able to increase total serum L-carnitine levels while simultaneously decreasing the characteristic rise in CK compared to the PLA. These results agreed with previous short- and medium-term studies, which also found that L-carnitine was able to reduce other markers of muscle damage, such as cytosolic serum proteins [5,11]. Volek and colleagues [5] found that 3 g per day of L-carnitine tartrate supplementation, providing 2 g of elemental L-carnitine for a period of 3 weeks, was able to attenuate biochemical and structural stress responses, and promote greater perceived recovery from muscle soreness and lower serum CK levels following a high–repetition squat protocol (5 sets, 15–20 repetitions using a load that was 50% of the subject’s squat one repetition maximum). Collectively, the evidence supports that L-carnitine supplementation can aid in recovery from muscle damage by decreasing blood markers of exercise.

In this study, recovery after moderate exercise was used as a measure of fatigue. A primary component necessary for successful exercise adherence is the level of recovery attained before initiating subsequent bouts of exercise [17,18]. Despite the potential value and importance of monitoring individuals’ recovery status, there are few practical options that are adequate or convenient for monitoring day-to-day recovery and fatigue [17]. One such tool is the PRS, which was validated as an indicator of recovery, soreness, fatigue, and readiness to train [18]. A previous study demonstrated that PRS scores are directly related to performance and indirectly related to soreness [17]. Moreover, elevations in perceived exercise-related soreness were shown to lower training adherence [23,24]. As such, ingredients that can improve perceptual outcomes may be highly beneficial. Our results demonstrated that L-carnitine tartrate supplementation blunted the decline in normalized PRS scores that were exhibited by the PLA group. These findings agreed with previous studies, which found that L-carnitine tartrate supplementation favorably improved perceptual measures of recovery [5,25].

Muscle strength and power are two of the most critical attributes underlying general health and longevity, particularly in the aging population [26]. These variables are intimately related and allow individuals to optimally perform activities of daily living and sustain long-term independence [26]. Power was assessed through both the SJ and CMJ, while strength was assessed via an IMTP. Collectively, these measures have been shown to relate to a multitude of functional and performance outcomes [27]. The relative delta changes in all three anaerobic outcome variables demonstrated greater decrements in the PLA compared to the CAR group, indicating that L-carnitine tartrate is effective at reducing the deleterious effects that intense exercise can have on both strength and power. Previous pilot research from Walter et al. [28] addressed the impact of L-carnitine on power output during a recovery period following a strenuous workout. Recovery of power was increased in 9 out of the 12 subjects receiving 2 g of L-carnitine daily over a period of 5 days. Moreover, a single administration of L-carnitine before exhaustive cycling exercise did not improve performance during a second round of exercise after 3 h [28]. In conjunction, the data from these studies, and the current study, indicate that L-carnitine supplementation speeds recovery of strength and power more consistently after supplementation for 5 weeks, while acute L-carnitine supplementation yielded inconclusive results on recovery of strength and power.

Gender-based differences in the physiological responses to exercise have been studied extensively for the last four decades, and yet the study of gender-specific dietary supplement outcomes on recovery has only been developing more in recent years [29,30]. Research has uncovered some specificity in females’ physiological response to exercise and determined that gender is an important variable to control for in the design of robust research protocols. For example, one study indicated that the anaerobic power and strength of females are lower than that of males [31]. This can be accounted for by differences in muscle- and fat-mass and anabolic hormone status [31]. Moreover, evidence indicating that neuromuscular fatigue during strenuous loading may be greater, and recovery from fatigue may be slower, in males compared to females, was given [32,33]. Research has shown that trained females can greatly outperform non-trained males in many circumstances [1]. Within this context, a growing number of commentaries have emphasized the need to study the effects of nutritional supplements on gender-specific recovery in sport, particularly among female athletes where strong scientific data are lacking [34]. Therefore, an important exploration of this study was to conduct a gender-specific sub-analysis. We found that L-carnitine tartrate lowered muscle damage, and accelerated recovery of anaerobic performance, in a similar pattern in male and female subjects. These results agreed with Ho et al. [12] who also found that L-carnitine tartrate was able to improve recovery in a gender-independent manner. Collectively, these findings indicate that L-carnitine tartrate can improve recovery in both males and females.

Aging is a physiological process that includes a gradual decrease in skeletal muscle mass, strength, and endurance, coupled with an ineffective response to tissue damage [35]. Decreases in the protein synthesis rate is affected by the translational process occurring in older human skeletal muscle. After muscle damage, muscles in older subjects do not regenerate as well compared to young adults [35]. The decreased regeneration capacity of muscles is due to both extrinsic (low amino acid intake and lack of muscle stimulation due to low physical activity) and intrinsic factors (blunted protein synthesis responses, and increased circulation of inflammatory and catabolic cytokines) [36]. Moreover, the degradation rate of contractile proteins in skeletal muscle during aging increases by about two times, and muscle strength and motor activity decrease concurrently [36]. Therefore, an important consideration of this study was to conduct an age-specific sub-analysis. Generally speaking, we found that L-carnitine tartrate supplementation lowered muscle damage and accelerated recovery in middle-aged and older adults. Previous research corroborates the results of this study, namely that L-carnitine tartrate was able to improve recovery in middle-aged individuals [12]. Collectively, these findings indicate that L-carnitine tartrate can improve recovery in an older adult population.

Since improving recovery by lowering muscle fatigue is of great interest to active individuals and athletes alike, it is worth exploring how L-carnitine tartrate may influence these outcomes in the general population. Reactive oxygen species (ROS) production following exercise-induced muscle damage was suggested [37]. Briefly, exercise elevates metabolism, and the use of oxygen is heightened [37]. The result is a leakage of ROS from the mitochondria [38]. Superoxide is a major ROS produced as a by-product of oxygen metabolism and, if not neutralized, can cause a host of cellular damage [38]. As such, ROS alters cell structure and cell function, and may contribute to muscle damage, leading to declines in performance [39]. The current study demonstrated that exercise impacted biochemical (e.g., serum CK), perceptual (e.g., PRS) and functional (strength and power) indices of muscle damage, fatigue, and recovery. The current trial showed that muscle damage after exercise was lowered by L-carnitine tartrate supplementation, most likely by lowering ROS, as evidenced by an increase in serum SOD, a major cellular antioxidant enzyme. Previous research indicated that 6 weeks of L-carnitine tartrate supplementation increased serum SOD levels, improving the antioxidant status in an animal model [7]. Therefore, we hypothesized and confirmed, in the present trial, that L-carnitine tartrate supplementation for 5 weeks also increased serum SOD levels in both males and females, as well as across age groups. In addition to its effects on improving antioxidant status, the present study showed that L-carnitine tartrate supplementation can also decrease exercise-induced inflammation, as evidenced by a decrease in serum CRP. Consequently, it is possible that the supplemented ingredient operates through improving antioxidant status and anti-inflammatory parameters, hence decreasing muscle damage and fatigue.

Interestingly, we found that females, during their menstruation cycle, had decreased levels of serum carnitine compared to pre-menstruation. L-carnitine supplementation was found to blunt this decrease and normalize these levels. It was reported that female athletes perform less during their menstruation cycle due to lack of energy, among other factors [40]. Normalized serum L-carnitine, which is involved in the generation of energy, may be suggested as means of helping to improve performance. Based on the limited dataset (19 females), we were not able to see differences in muscle power or strength.

## 5. Conclusions

In conclusion, our trial demonstrated that L-carnitine tartrate supplementation, over a period of 5 weeks, was able to improve recovery and fatigue based on reduced muscle damage and soreness after an exercise challenge in a relatively large cohort of male and female subjects, aged 21- to 65-years. Muscle power and strength were also improved in both males and females, independently of age. Stratification analysis based on gender showed that L-carnitine can help to support exercise recovery needs, particularly among females. These findings may support practitioners’ decisions in recommending supplementation for athletes, of different genders and among the general population, seeking to adhere to various exercise protocols.

## Figures and Tables

**Figure 1 nutrients-13-03432-f001:**
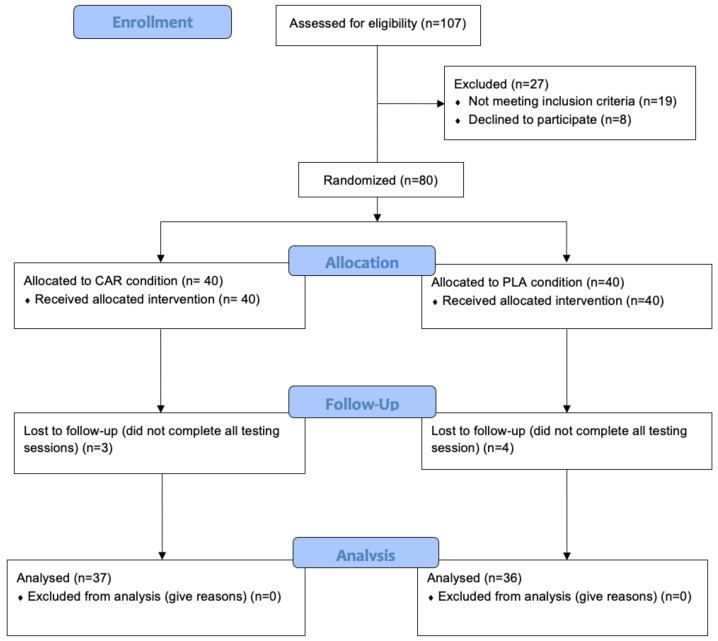
CONSORT flow diagram. CAR: L-carnitine group; PLA: placebo.

**Figure 2 nutrients-13-03432-f002:**
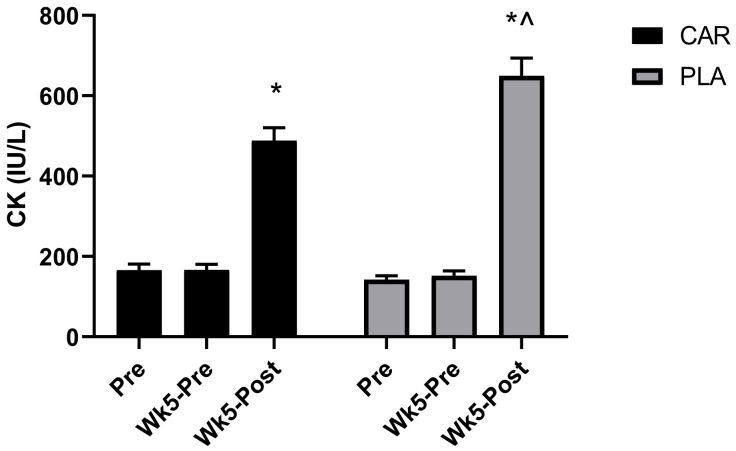
Serum creatine kinase (CK) concentration. Mean values and error bars for the standard error of the mean for concentration values of serum creatine kinase (CK) are shown. * indicates a significant difference between the time point Pre and Wk5-Post (*p* < 0.001). ^ indicates a significant difference between groups (*p* < 0.05). The groups are carnitine (CAR) and placebo (PLA).

**Figure 3 nutrients-13-03432-f003:**
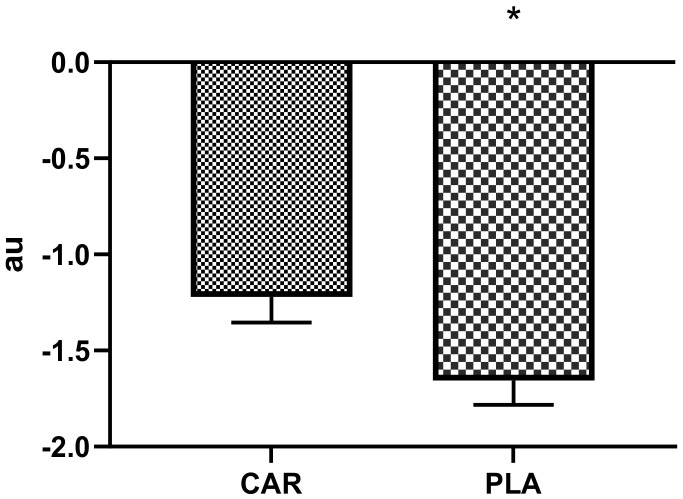
Relative perceived recovery status. Mean values and error bars for the standard error of the mean for perceived recovery values normalized to training volume for the muscle damaging protocol are shown. The following equation was deployed to reach the arbitrary units (au): (Post Exercise PRS - True Baseline Pre)/total repetitions) × 100; * indicates a significant difference between groups (*p* < 0.05). The groups are carnitine (CAR) and placebo (PLA). Abbreviations: au—arbitrary units.

**Figure 4 nutrients-13-03432-f004:**
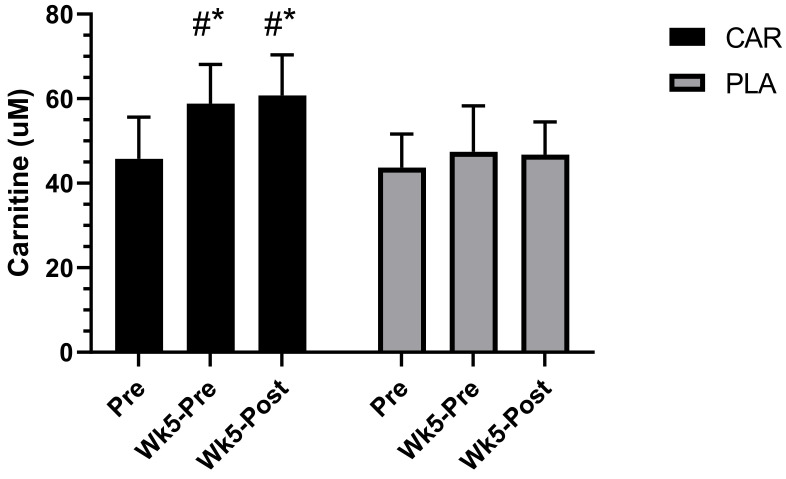
Serum carnitine concentration. Mean values and error bars for the standard error of the mean for concentration values of total serum carnitine concentration are shown. # indicates a significant difference from the time point Pre (*p* < 0.001). * indicates a significant difference between groups (*p* < 0.001). The groups are carnitine (CAR) and placebo (PLA).

**Figure 5 nutrients-13-03432-f005:**
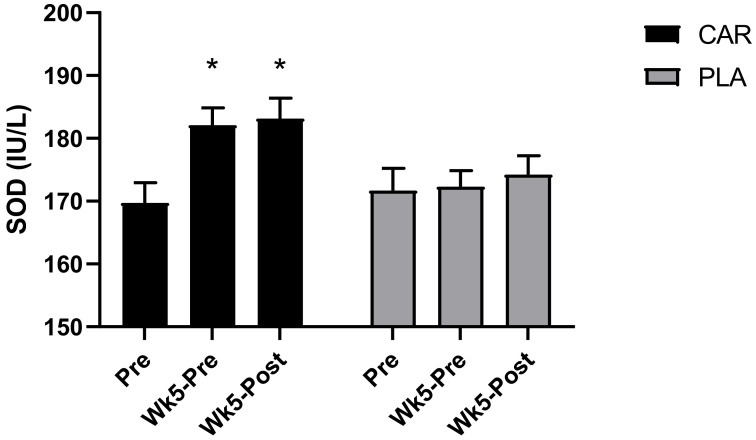
Serum superoxide dismutase (SOD). Bar chart for the mean values and error bars for the standard error of the mean for serum super oxide dismutase (SOD) are shown. * indicates a significant difference from Pre (*p* < 0.001). The groups are carnitine (CAR) and placebo (PLA).

**Figure 6 nutrients-13-03432-f006:**
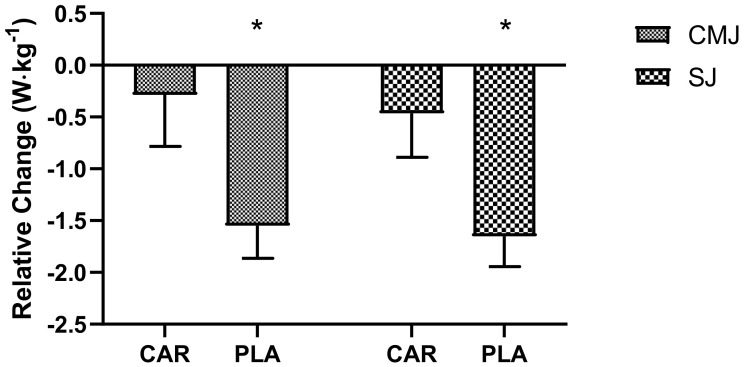
Maximum relative power output delta. Mean values and error bars for the standard error of the mean for the relative delta change (Time_2_ − Time_1_), from Wk5-Pre to Wk5-Post, of power output from the countermovement jump (CMJ) and squat jump (SJ) are shown; * indicates a significant difference between groups (*p* < 0.05). The groups are carnitine (CAR) and placebo (PLA).

**Figure 7 nutrients-13-03432-f007:**
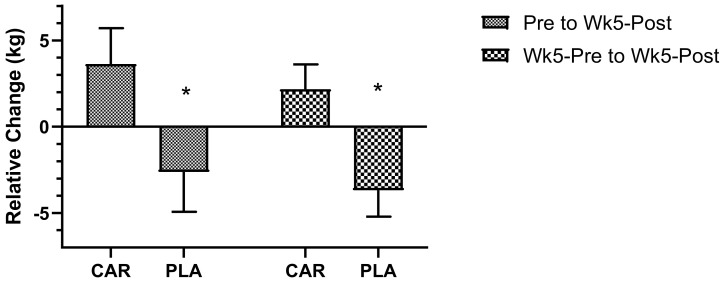
Maximal isometric strength delta. Mean values and error bars for the standard error of the mean for the relative delta change (Time_2_ - Time_1_), from Pre to Wk5-Post and Wk5-Pre to Wk5-Post, of strength output of the isometric midthigh pull are shown; * indicates a significant difference between groups (*p* < 0.05). The groups are carnitine (CAR) and placebo (PLA).

**Figure 8 nutrients-13-03432-f008:**
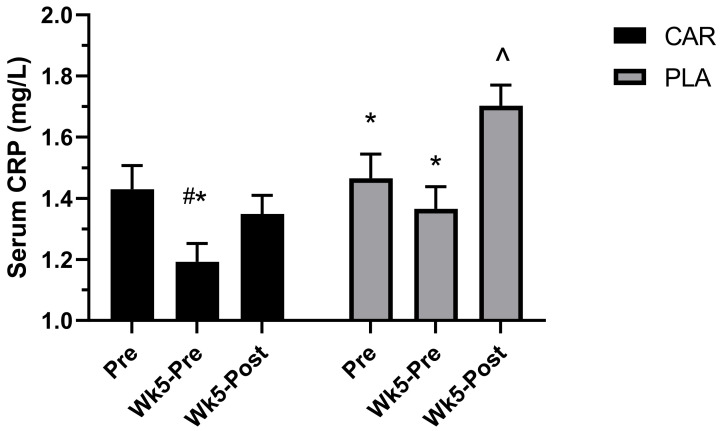
Serum C-reactive protein (CRP). Mean values and error bars for the standard error of the mean of serum C-reactive protein (CRP) levels are shown. # indicates a significant difference from Pre (*p* < 0.001). * indicates a significant difference from Wk5-Post (*p* < 0.005). ^ indicates a significant difference between groups (*p* < 0.001). The groups are carnitine (CAR) and placebo (PLA) [21].

**Figure 9 nutrients-13-03432-f009:**
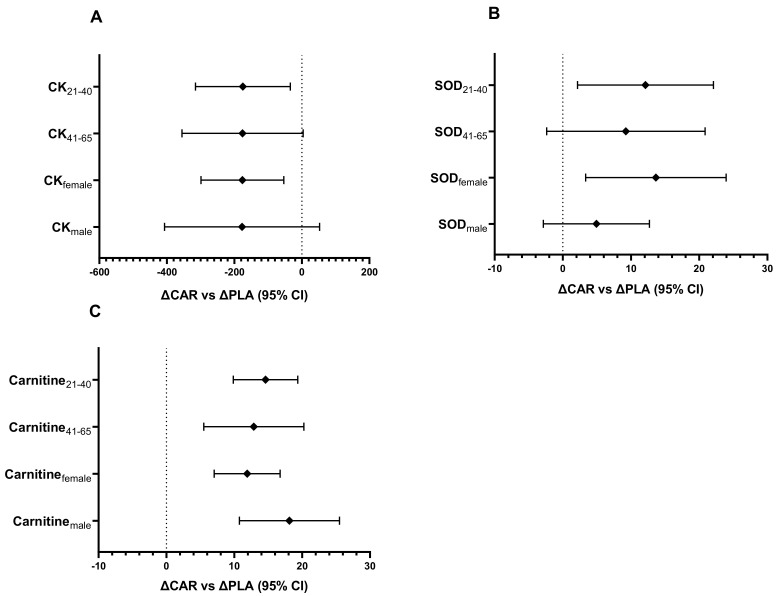
CK (**A**), SOD (**B**), and carnitine (**C**) ΔCAR vs. ΔPLA (95% CI). The 95% confidence intervals of the between group difference for the relative change from Wk5-Pre to Wk5-Post (∆CAR vs. ∆PLA) by age and gender groups for CK (**A**), and the between-group difference for the relative change from Pre to Wk5-Post (∆CAR vs. ∆PLA) by gender and age groups for SOD (**B**) and carnitine (**C**) are shown. The groups are carnitine (CAR) and placebo (PLA). Abbreviations: CK = creatine kinase; SOD = superoxide dismutase.

**Figure 10 nutrients-13-03432-f010:**
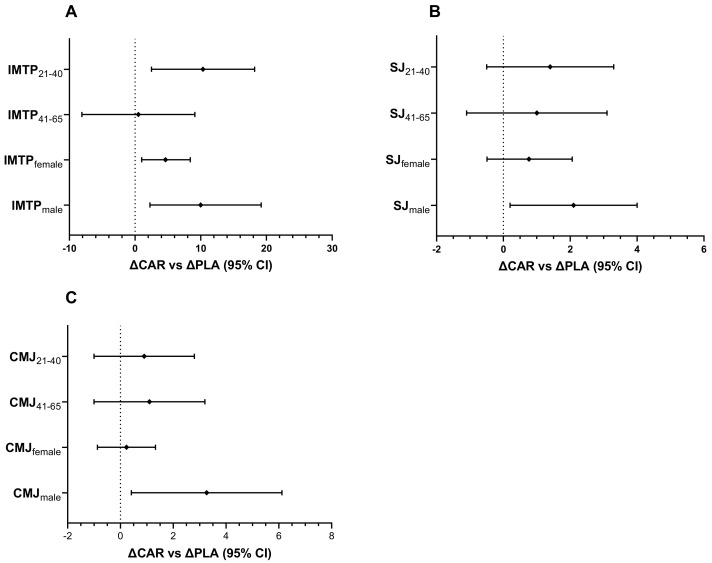
Performance metrics ΔCAR vs. ΔPLA (95% CI). The 95% confidence intervals of the between-group difference are shown for the relative change from Wk5-Pre to Wk5-Post (∆CAR vs. ∆PLA) by gender and age for IMTP (**A**), SJ (**B**), and CMJ (**C**). The groups are carnitine (CAR) and placebo (PLA). Abbreviations: IMTP = isometric mid-thigh pull; SJ = squat jump; CMJ = countermovement jump.

**Figure 11 nutrients-13-03432-f011:**
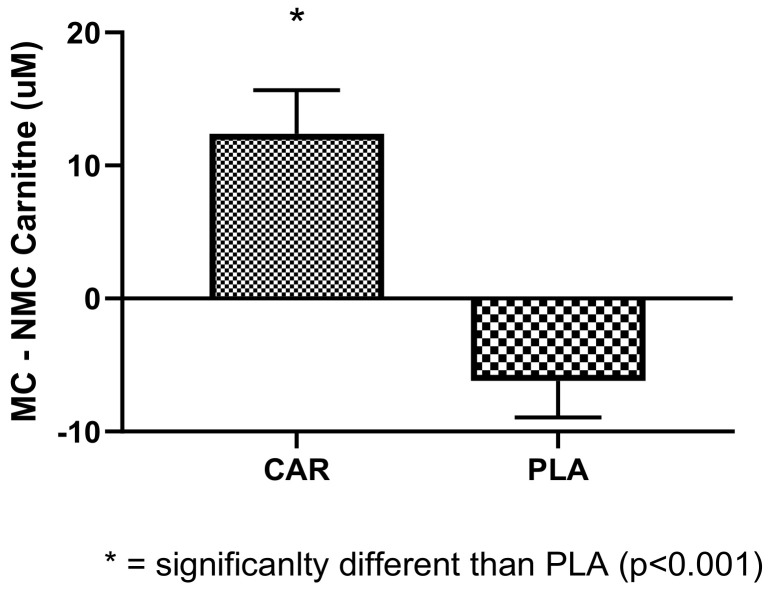
Changes in serum carnitine levels during the menstruation cycle from pre-menstruation are shown. * indicates statistical significance between groups (*p* < 0.001). The groups are carnitine (CAR) and placebo (PLA). Abbreviations: MC = menstrual cycle; NMC = non-menstrual cycle.

**Table 1 nutrients-13-03432-t001:** Baseline Subject Characteristics.

	CAR			PLA		
	Males	Females	Total	Males	Females	Total
Sample Size	12	25	37	10	26	36
Age (years)	39.3 ± 1.3	40.5 ± 1.5	40.1 ± 1.1	34.0 ± 2.4	40.6 ± 2.2	38.8 ± 1.8
Height (cm)	177.8 ± 1.1	163.4 ± 1.3	168.1 ± 1.5	180.1 ± 2.7	165.6 ± 1.8	169.6 ± 1.8
Body Mass (kg)	87.8 ± 3.5	64.1 ± 1.1	71.8 ± 2.3	97.0 ± 5.1	63.9 ± 1.2	73.1 ± 3.0
Body Fat (%)	23.1 ± 1.5	32.4 ± 1.3	29.4 ± 1.2	24.4 ± 1.2	30.7 ± 0.8	28.9 ± 0.8
BMI (kg/m^2^)	27.7 ± 3.2	24.1 ± 2.4	25.3 ± 3.2	29.8 ± 3.2	23.5 ± 3.4	25.25 ± 4.3
Ethnicity—*n* (%)						
White	11 (92%)	22 (88%)	33 (89%)	7 (70%)	23 (88%)	30 (83%)
Hispanic	1 (8%)	3 (12%)	4 (11%)	1 (10%)	2 (8%)	3 (8%)
Black	0 (0%)	0 (0%)	0 (0%)	2 (20%)	1 (4%)	3 (8%)

Data are mean ± SEM. CAR: L-carnitine group; PLA: placebo.

**Table 2 nutrients-13-03432-t002:** Adverse Events Report.

Adverse Event	CAR	PLA
Swallowing	3	3
Nausea	6	7
Upset Stomach	6	6
Headache	11	10
Cramping	16	14
Total	42	40

## Data Availability

Data described in the manuscript will be made available upon request by the corresponding author.

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
