# Peer review of "L-Carnitine Tartrate Supplementation for 5 Weeks Improves Exercise Recovery in Men and Women: A Randomized, Double-Blind, Placebo-Controlled Trial"

_nutrients, 2021, doi:10.3390/nu13103432_

Round 1

Reviewer 1 Report

The authors investigated the effects of daily L-carnitine tartrate supplementation for 5 weeks on recovery and fatigue of healthy adults. The manuscript has some good information for readers and the scientific community. The reviewer has some comments/concerns and recommended major revisions.

Specific points:

A 5-week study is not considered a long-term or chronic study. Please avoid using “long-term” or “chronic” through the manuscript.

Table 1: please add BMI and race information for each group.

Line 135: Was fasting venous blood collected? How long did the subjects fast?

Lines 274-298: How did you estimate the sample size? Will current sample size provide enough power for the primary outcome? What did you do if data were not normally distributed? Why some outcome variables were analyzed using ANOVA, but some were analyzed using relative delta change and t-test but not ANOVA? Which software was used for statistical analysis? Gender and age should be included as covariate if they have significant effect.

Line 307: Does 194% mean “increased by 194%”?

Figure 2: The Figure 2 used *, but the figure legend annotated #.

Line 335: please check the unit of serum carnitine. In the text, mM was used, while the Figure 4 showed uM.

Line 346-350: Please clarify if between-group difference of SOD was observed.

Figure 6: The column fill pattern for CMJ and SJ are same.

Lines 416-470: For the performance metrics stratified by age and gender, it may be interesting to understand if there is any significant gender or gender*treatment, and age or age*treatment effects. However, the authors only compared CAR vs PLA within each stratified group.

Author Response

Reviewer #1

The authors investigated the effects of daily L-carnitine tartrate supplementation for 5 weeks on recovery and fatigue of healthy adults. The manuscript has some good information for readers and the scientific community. The reviewer has some comments/concerns and recommended major revisions.

Specific points:

Comment: A 5-week study is not considered a long-term or chronic study. Please avoid using “long-term” or “chronic” through the manuscript.

Response: We would like to thank the reviewer for the comment and we did change the language that included “long-      term” or “chronic” through the text (Tracked in the text).

Comment: Table 1: please add BMI and race information for each group.

Response: Thank you for this suggestion. BMI and race information for each group was added to Table 1.

Comment: Line 135: Was fasting venous blood collected? How long did the subjects fast?

Response: Yes it was fasting venous blood after a 10h fast. This has been     addressed and updated to’ “Subjects were 10h fasted prior to venous blood being extracted”. Line 155

Comment: Lines 274-298: How did you estimate the sample size? Will current sample size provide enough power for the primary outcome? What did you do if data were not normally distributed? Why some outcome variables were analyzed using ANOVA, but some were analyzed using relative delta change and t-test but not ANOVA? Which software was used for statistical analysis? Gender and age should be included as covariate if they have significant effect.

Lines 416-470: For the performance metrics stratified by age and gender, it may be interesting to understand if there is any significant gender or gender*treatment, and age or age*treatment effects. However, the authors only compared CAR vs PLA within each stratified group.

Response: A formal power analysis was not done for sample size estimation in this study. The 80 subjects enrolled with 73 completers was based on a regulatory requirement for a health claim request in MFDS Korea. However, previous studies (references listed below and included in the manuscript) with L-carnitine and similar endpoints enrolled lower number of subjects. In Volek et al. (2004) the sample size was n=17, Ho et al. (2010) n=18, Koozehchian et al. (2018) n=23. In addition, a recent meta-analysis on the effect of L-carnitine supplementation on exercise-induced muscle damage by Yarizadh et al. (2020) includes nine additional studies (Table 1, Yarizadh et al. (2020)) all with sample sizes significantly lower than 80. The authors feel that a total sample size of 73 gives enough power to the study particularly compared to the described literature. In fact, we were able to conduct a post hoc computation of achieved power using creatine kinase (primary outcome) which indicates that power (1 – β error probability) was 0.886. This example considered the group’s mean difference from Wk5-Pre to Wk-5-Post (CAR: mean = 322.1, SD =             204.6; PLA: mean = 497.6, SD = 259.5), which yields an effect size of d=0.751 ((M2 - M1) ⁄ SD pooled). Using G*Power (version 3.0.10), the following settings were applied to return an achieved power of 0.886:

Test family: t test

Statistical test: difference between two independent means

Type of power analysis: compute achieved power given alpha, sample size, and effect size

Tails: two

Effect size d: 0.751 (computed in G*power and transferred to input window)

Alpha err prob: 0.05

Sample size group 1 (CAR): 37

Sample size group 2 (PLA): 36

Normality was graphically assessed through visual inspections of Q-Q plots and then confirmed through Shapiro-Wilk testing (p>0.05). Normality was not violated for the dependent variables accessed in this study, therefore non-parametric alternatives nor log transformations were used. Statistical analyses were completed using GraphPad Prism 9; this has been added to the manuscript. Gender and age were not included as covariates because they did not impact the model. For all primary (section 3.1) and secondary objectives (3.2), ANOVA outcomes are reported in text, even if they were also analyzed with t-test. With absence of a significant group x time interaction, the ANOVA model does not permit the decomposing of factors to assess group differences. Hence, we utilized t-test on the group’s relative delta change. The authors believe that significant changes in means (t-test) is relevant information despite the lack of group x time interaction (ANOVA). Due to the extensive nature of our exploratory stratification analyses we opted to focus on relative changes rather than gender and/or age x treatment effects because we did not want the readers to confuse gender x treatment or age x treatment effects with the significant interaction effects in primary and secondary objectives.

References

Ho, J.-Y., Kraemer, W. J., Volek, J. S., Fragala, M. S., Thomas, G. A., Dunn-Lewis, C., … Maresh, C. M. (2010). l-Carnitine l-tartrate supplementation favorably affects biochemical markers of recovery from physical exertion in middle-aged men and women. Metabolism, 59(8), 1190–1199.

Koozehchian, M. S., Daneshfar, A., Fallah, E., Agha-Alinejad, H., Samadi, M., & Kaviani, M. (2018). Effects of nine weeks L-Carnitine supplementation on exercise performance, anaerobic power, and exercise-induced oxidative stress in resistance-trained males. Journal of exercise nutrition & biochemistry, 22(4), 7.

Volek, J. S., Ratamess, N. A., Rubin, M. R., Gomez, A. L., French, D. N., McGuigan, M. M., ... & Kraemer, W. J. (2004). The effects of creatine supplementation on muscular performance and body composition responses to short-term resistance training overreaching. European journal of applied physiology, 91(5), 628-637.

Yarizadh, H., Shab-Bidar, S., Zamani, B., Vanani, A. N., Baharlooi, H., & Djafarian, K. (2020). The effect of l-carnitine supplementation on exercise-induced muscle damage: a systematic review and meta-analysis of randomized clinical trials. Journal of the American College of Nutrition, 39(5), 457-468.

Comment: Line 307: Does 194% mean “increased by 194%”?

Response: Yes the increase is by 194 % in the CAR group and 351% in PLA however the difference between groups is significant. For clarification, the line 331 has been amended and “greater increase in CK” has been added.

Comment: Figure 2: The Figure 2 used *, but the figure legend annotated #.

Response: The annotation was changed from # to * In addition, Pre to Wk5-Pre was corrected to Pre to Wk5-Post.

Comment: Line 335: please check the unit of serum carnitine. In the text, mM was used, while the Figure 4 showed uM.

Response: The correct unit for carnitine in this study is uM. It has been updated in the text to reflect that.

Comment: Line 346-350: Please clarify if between-group difference of SOD was observed.

Response: This has been clarified, and “no difference between groups was detected” has been added.

Comment: Figure 6: The column fill pattern for CMJ and SJ are same.

Response: Upon close examination, the CMJ and SJ column fill pattern is not the same. Therefore we kept the fill pattern as it is.

Reviewer 2 Report

I comment on “L-Carnitine Tartrate Supplementation Improves Long-Term Recovery in Men and Women: A Randomized, Double-Blind, Placebo-Controlled Trial” an original article aiming to evaluate the effects of daily L-carnitine tartrate supplementation for 5 weeks on recovery and fatigue.

Abstract is well structured and provides all the necessary information to attract readership’s attention.

Introduction’s first paragraphs are concise and provide a general picture of L-carnitine, its’ production, intake and interactions in human body (muscles’ damage attenuation). The last paragraph presents clear the aim of this study, which is the investigation of the long-term effects.

Material and Methods were thoroughly described. Subjects were isolated based on several exclusion factors. It is described in details. I would like to know more about the activity level classification. Maybe it would interest some readers too (you could add more information). The registration of study is valid. I have no further comments to make as everything else is painstakingly executed and described herein.

Results are presented clearly and with all the necessary figures and graphs.  

Discussion certainly caught my interest.

Author Response

Reviewer #2

Comment: I comment on “L-Carnitine Tartrate Supplementation Improves Long-Term Recovery in Men and Women: A Randomized, Double-Blind, Placebo-Controlled Trial” an original article aiming to evaluate the effects of daily L-carnitine tartrate supplementation for 5 weeks on recovery and fatigue.

Abstract is well structured and provides all the necessary information to attract readership’s attention.

Introduction’s first paragraphs are concise and provide a general picture of L-carnitine, its’ production, intake and interactions in human body (muscles’ damage attenuation). The last paragraph presents clear the aim of this study, which is the investigation of the long-term effects.

Material and Methods were thoroughly described. Subjects were isolated based on several exclusion factors. It is described in details. I would like to know more about the activity level classification. Maybe it would interest some readers too (you could add more information). The registration of study is valid. I have no further comments to make as everything else is painstakingly executed and described herein.

Results are presented clearly and with all the necessary figures and graphs.  

Discussion certainly caught my interest.

Response: Thank you for all of your kind comments. The activity level for the subjects in the methods section was added (lines 91-93) to provide more clarity and detail.

Round 2

Reviewer 1 Report

The authors have addressed most of my concerns and the revised manuscript looks better. However, I still have a concern on the Subanalysis of Serum CK, SOD, and Carnitine Levels Stratified by Age and gender. The authors mentioned that gender and age didn’t impact the model, which means gender and age didn’t significantly affect the results. In this case, it is meaningless to do subanalysis stratified by age and gender.

Author Response

Reviewer #1 round 2

Comments and Suggestions for Authors

The authors have addressed most of my concerns and the revised manuscript looks better. However, I still have a concern on the Subanalysis of Serum CK, SOD, and Carnitine Levels Stratified by Age and gender. The authors mentioned that gender and age didn’t impact the model, which means gender and age didn’t significantly affect the results. In this case, it is meaningless to do subanalysis stratified by age and gender.

Response: We thank you the reviewer for the comment and we did look at the significance based on three-way interactions or two-way interactions for age/gender x group. We did not see differences, suggesting that these results are independent of age and gender. We think it is important to keep this information as this suggests that the observed benefits are not driven by a gender or an age group. For clarification we added the following sentence at the end of this result section in line 487 “This analysis indicate that the effects seen in CK, SOD and serum carnitine are significant across ages and in both gender.